# Identification of the Volatile Components of *Galium verum* L. and *Cruciata leavipes* Opiz from the Western Italian Alps

**DOI:** 10.3390/molecules25102333

**Published:** 2020-05-16

**Authors:** Aldo Tava, Elisa Biazzi, Domenico Ronga, Pinarosa Avato

**Affiliations:** 1CREA Research Centre for Animal Production and Aquaculture, viale Piacenza 29, 26900 Lodi, Italy; elisa.biazzi@crea.gov.it (E.B.); d.ronga@crpa.it (D.R.); 2Centro Ricerche Produzioni Animali—CRPA S.p.A., viale Timavo, n. 43/2, 42121 Reggio Emilia, Italy; 3Dipartimento di Farmacia-Scienze del Farmaco, Università, via Orabona 4, 70125 Bari, Italy; pinarosa.avato@uniba.it

**Keywords:** *Cruciata laevipes* Opiz, *Galium verum* L., essential oil composition, GC/FID, GC/MS, *Rubiaceae*, benzyl alcohol, β-caryophyllene, methylbenzaldehyde, phenylacetaldehyde, VOCs, Alpine plants

## Abstract

The chemical composition of the volatile fraction from *Galium verum* L. (leaves and flowers) and *Cruciata laevipes* Opiz (whole plant), *Rubiaceae*, was investigated. Samples from these two plant species were collected at full bloom in Val di Susa (Western Alps, Turin, Italy), distilled in a Clevenger-type apparatus, and analyzed by GC/FID and GC/MS. A total of more than 70 compounds were identified, making up 92%–98% of the total oil. Chemical investigation of their essential oils indicated a quite different composition between *G. verum* and *C. laevipes*, both in terms of the major constituents and the dominant chemical classes of the specialized metabolites. The most abundant compounds identified in the essential oils from *G. verum* were 2-methylbenzaldheyde (26.27%, corresponding to 11.59 μg/g of fresh plant material) in the leaves and germacrene D (27.70%; 61.63 μg/g) in the flowers. *C. laevipes* essential oils were instead characterized by two sesquiterpenes, namely β-caryophyllene (19.90%; 15.68 μg/g) and *trans*-muurola-4(15),5-diene (7.60%; 5.99 μg/g); two phenylpropanoids, benzyl alcohol (8.30%; 6.71 μg/g), and phenylacetaldehyde (7.74%; 6.26 μg/g); and the green-leaf alcohol *cis*-3-hexen-1-ol (9.69%; 7.84 μg/g). The ecological significance of the presence of such compounds is discussed.

## 1. Introduction

*Galium verum* L. and *Cruciata laevipes* Opiz (syn. *Galium cruciata* (L.) Scop.) belong to the Rubiaceae plant family, namely to the Rubieae monophyletic group, the only tribe classified within the family. Phylogenetic studies [1,2] showed that the tribe can be separated into two groups, one of them including both genera, *Galium*, the largest genus within the tribe with 655 species, and *Cruciata* including only 9 species. In addition, based on DNA sequence data, it was demonstrated that the two species *G. verum* and *C. laevipes* fall into two separated clades, whose members are also characterized by some morphological differences.

*G. verum* L., is a herbaceous perennial species largely spread across most of Europe, North Africa, and temperate Asia. It grows between 0 and 1800 m above the sea level in different habitats, including dry-sand meadows, rocky outcrops, roadsides, dunes, and seashores. *G. verum* is a scrambling plant, with 60–120 cm long stems that root frequently when they touch the soil. The leaves are linear needle-like, shiny dark green, with prominently revolute margins, covered with hair underneath, grouped in whorls of eight–twelve. It blooms in July–September producing fragrant yellow flowers, clustered in dense panicles. The fruits are black schizocarps [3].

The name of the genus derives from the Greek term “*gala*”, milk, referring to the common use of plants of this species to curdle milk when producing cheese. *G. verum* is well-known by the common name of lady’s bedstraw or yellow spring bedstraw (madder family), from the old practice to use its foliage to stuff mattresses. In addition, the flowers give a yellow pigment that has been employed traditionally to color food, and the roots produce a red dye that is used to color wool. The plant also has some traditional medical uses as a diuretic, choleretic, and spasmolytic [4].

*C. laevipes* Opiz, commonly known as crossword or smooth bedstraw, is also distributed across Europe and Asia, growing between 0 and 1500 m above the sea level. It is a perennial herb, 10 to 60 cm tall; with quadrangular stems; and leaves that are three-nerved and whorled in clusters of four. The flowers are yellow, between five and nine on each top, blooming in May–June; their peduncles carry two bracteoles each and are filled with dense hairs [3]. This species grows in open woodland, waysides, and pastures, and prefers calcareous soils [5]. *C. laevipes* has long been known in folk medicine for its wound-healing properties, and it was employed in the past in both external and internal applications. These latter include remedies to treat obstructions of the stomach and bowels, to stimulate appetite and as a remedy for rheumatism and dropsy [6].

The two plant species, *G. verum* and *C. laevipes* are also widespread in Italy, all over the country, especially in the Alpine regions, and were also commonly found in the Western Italian Alpine pasture vegetation [7].

Phytochemical studies have shown that both *Galium* and *Cruciata* genera synthesize many different classes of specialized metabolites, such as iridoid glycosides, antraquinones, phenolics, flavonoids, and coumarins [8,9,10,11,12,13,14,15,16,17], possibly accounting for the biological properties highlighted for some of the species. They also produce terpenoids and aromatic essential oils [18,19,20,21].

The aim of this work was to determine the flavor constituents of *G. verum* L. and *C. laevipes* Opiz collected in the wild Italian alpine region, in order to fully characterize their volatile fraction for the first time.

## 2. Results

Volatiles identified in the aerial parts of *G. verum* and *C. laevipes* are reported in Table 1, listed in order of elution on a DB-5 column. Leaves and flowers from *G. verum* were analyzed separately, while, due to their small size, flowers from *C. laevipes* could not be isolated and for this species the whole plant was analyzed.

In total, more than 70 compounds were identified in both species, on average amounting to 92%–98% of the total. Chemical investigations of their essential oils indicated a quite different composition between *G. verum* and *C. laevipes,* both in terms of major constituents and the dominant chemical classes of the specialized metabolites (Table 1; Figure 1). The chemical structures of the most abundant detected compounds are reported in Figure 2.

Aldhehydes were the most abundant chemical class of components amounting to 43.71 ± 0.01% (18.91 ± 0.44 μg/g) and to 42.64 ± 0.46% (94.93 ± 2.33 μg/g) in *G. verum* essential oils from leaves and flowers, respectively. The second major chemical class was represented by alcohols in the leaves (25.70 ± 1.67%, 11.11 ± 0.47 μg/g) and by terpenes in the flowers (32.42 ± 1.24%, 72.15 ± 1.77 μg/g). Alcohols were also present in high amount (12.09 ± 0.24%, 26.92 ± 0.89 μg/g) in the essential oils from the flowers of *G. verum* (Table 1).

2-Methylbenzaldehyde was the main component identified in the essential oils obtained from *G. verum* leaves, accounting for 26.27 ± 1.07% (11.59 ± 0.73 μg/g). This aldehyde also represented a dominant constituent in essential oils from the flowers of the same species (24.04 ± 1.07%, 53.54 ± 3.12 μg/g). A second major aldehyde detected in the essential oils from these two plant organs from *G. verum* was represented by 4-methylbenzaldehyde (Table 1). The second major component characteristic of the essential oils from *G. verum* leaves was *cis*-3-hexen-1-ol, amounting to 17.34 ± 2.41% (7.49 ± 0.87 μg/g).

The sesquiterpene germacrene D was the main metabolite (27.70 ± 1.67%, 61.63 ± 2.87 μg/g) identified in the flowers from essential oils of the same species.

Both samples of essential oils obtained from *G. verum* also contained small quantities of hydrocarbons and esters (Table 1), amounting, respectively, to 8.45 ± 0.22% (leaves) and 2.18 ± 0.25% (flowers), and to 3.76 ± 1.29% (leaves) and 1.52 ± 0.04% (flowers). Among these, a number of linear-chained alkanes such as nonacosane (2.98 ± 0.21%, 1.29 ± 0.12 μg/g), tricosane (1.62 ± 0.17%, 0.70 ± 0.06 μg/g), and eptacosane (0.72 ± 0.14%, 0.31 ± 0.05 μg/g) was especially abundant in the essential oils distilled from the leaves (Table 1).

The ester fraction contained, almost entirely, *cis*-3-hexenylacetate, accounting for 3.46 ± 1.30% (1.50 ± 0.60 μg/g) and 1.12 ± 0.02% (2.50 ± 0.04 μg/g) in the *G. verum* leaves and flowers, respectively.

*C. laevipes* essential oil yielded an interesting mixture of volatile compounds (Figure 1), with terpenes as the best represented chemical group, accounting for 46.11 ± 3.21% of the total oil, corresponding to 36.34 ± 3.21 μg/g fresh weight. Borneol was the most abundant monoterpene with 4.07 ± 0.43% (3.21 ± 0.32 μg/g) of the volatile fraction, while *β*-caryophyllene was the most abundant sesquiterpene and the single most abundant compound of the total oil, quoted as 19.90 ± 2.32% corresponding to 15.68 ± 1.92 μg/g fresh weight. Among other sesquiterpenes, *trans*-muurola-4(15),5-diene was detected at a relatively high amount, 7.60 ± 0.42% (5.99 ± 0.37 μg/g fresh weight), while *α*-humulene and the oxygenated terpene eudesma-4(15),7-dien-1*β*-ol were present at 2.51 ± 0.04% (1.98 ± 0.04 μg/g) and 2.60 ± 0.16% (2.05 ± 0.14 μg/g), respectively. Four oxygenated sesquiterpenes were also detected at percentages ranging between 0.70% and 1.20%, whose structure could not be identified. However, based on their MS fragmentation patterns, closely related to those of the standard compounds, it was possible to assign them the molecular formulas reported in Table 1.

Alcohols were the second most abundant chemical class of compounds, accounting for 22.72 ± 1.68% of the total volatiles, thus yielding 18.35 ± 1.85 μg/g fresh weight. Among them the aliphatic cis-3-hexen-1-ol showed the highest percentage value with 9.69 ± 1.18% (7.84 ± 1.17 μg/g), followed by the aromatic benzyl alcohol with 8.30 ± 0.24% (6.71 ± 0.24 μg/g).

Aldehydes accounted for 13.02 ± 0.86% of the total oil, corresponding to 10.50 ± 0.95 μg/g fresh weight. The two main compounds of this class were phenylacetaldehyde and benzaldehyde, accounting for 7.74 ± 0.41% (6.26 ± 0.52 μg/g) and 2.05 ± 0.21% (1.66 ± 0.21 μg/g) of the total volatiles.

Among other classes of compounds, acids accounted for 4.27 ± 0.04% of the total oil, corresponding to 3.37 ± 0.01 μg/g fresh weight, mainly represented by hexadecanoic acid (2.25 ± 0.36%, 1.98 ± 0.27 μg g^−1^). Phenolics were also present at detectable amounts, quantified as 3.97 ± 0.38% of the total oil (3.13 ± 0.28 μg/g fresh weight). The main component of this class was eugenol, which is well known as a natural antimicrobial agent [22], and is detected at 3.67 ± 0.39% of the total volatiles corresponding to 2.89 ± 0.29 μg/g fresh weight.

As in *G. verum* volatile oils, *C. laevipes* yielded a number of linear-chained alkanes, together accounting for 3.87 ± 0.11% of the oil and corresponding to 3.05 ± 0.10 μg/g fresh weight. Nonacosane (0.80 ± 0.05%, 0.63 ± 0.03 μg/g), tricosane (0.54 ± 0.04%, 0.43 ± 0.04 μg/g), and pentacosane (0.52 ± 0.08%, 0.41 ± 0.07 μg/g) were the most abundant homologues of this class.

Esters were also present in *C. laevipes* volatile fraction at 2.72 ± 0.29% (2.20 ± 0.29 μg/g fresh weight). Their presence is somehow significant since both compounds were detected, i.e., cis-3-hexenyl acetate (2.44 ± 0.27%, 1.97 ± 0.28 μg/g) and methyl salicylate (0.28 ± 0.02%, 0.22 ± 0.01 μg/g), which can both be associated to mechanisms of active plant defense [23,24].

Among the miscellaneous components worth mentioning was a relatively small amount of indole (0.34 ± 0.04%, 0.27 ± 0.03 μg/g), a metabolite that is possibly derived from the degradation of tryptophan, which is quite rare in plant volatiles and is associated with the presence of parasites in some cases [23].

## 3. Discussion

To the best of our knowledge, this is the first detailed investigation of the chemical composition of the essential oils produced by the two Rubiaceae species, *G. verum* and *C. leavipes*. In addition, this was the first characterization of these two species growing wild in the Italian Alpine environment.

Previous studies on the two species mainly dealt with the characterization of methanolic extractives [4,17]. The composition of volatiles obtained from the wild plants from East Europe was also reported [18,20] and in *G. verum*, only the flowers were analyzed. Data from the literature indicate a different chemical composition compared to our study, i.e., *G. verum* flowers were described to contain *cis*-3-hexen-1-ol as the most abundant component, followed by squalene [20]. Essential oils from *C. laevipes* were instead reported to produce borneol and verbenone, as the major terpenes [18].

Qualitative and quantitative differences can be possibly attributed to the different habitats in which the plant material used in our study was growing, i.e., the Italian Alpine environment.

Essential oils from the leaves of *G. verum* were characterized by a high amount of 2-methylbenzaldehyde, which is a compound that also naturally occurs in other aromatic plants such as Taraxacum officinale and Morinda officinalis [25,26], and was also reported as a component of the essential oils from *G. humifusum* [21]. This phytochemical and some derived molecules showed a strong anti-mite effect [26,27], thus suggesting its ecological contribution, and possibly of 4-methylbenzaldehyde, to prevent insect attacks.

On the other hand, essential oils distilled from *G. verum* leaves are very rich in cis-3-hexen-1-ol, and is well-known as a semiochemical acting as a repellant/attractant for herbivores [28].

The presence of germacrene D as the major metabolite in the flowers of the same species was also consistent with an ecological role. This sesquiterpene was reported to act as a pheromone with anti-herbivore properties and it has been reported to be repellent against aphids [29]. The same compound, however, often contributes to the floral scent of some plant species because of its importance as an attractant of pollinators [30].

With regards to *C. leavipes*, it should be underlined that the two major aromatic aldehydes, phenylacetaldeyde and benzaldehyde, largely exceeded the modest contribution of short-chained saturated and unsaturated aliphatic aldehydes (from C_6_ to C_10_). It is to be noted that most linear-chained aliphatic alcohols and aldehydes, also known as green-leaf volatiles, are derived from the enzymatic cleavage of C_18_ unsaturated acids, and play a major role in plant signaling and defense mechanisms [23,31]. Consequently, it is worth noting the presence of linolenic acid as one of the precursors of green-leaf volatiles, in both *G. verum* and *C. laevipes* essential oils [32,33]. On the other hand, aromatic alcohols and aldehydes are synthesized through the phenylpropanoid pathway, together with other benzenoids and phenolics, and they can be enzymatically converted into one another through specific dehydrogenases [34]. Since aromatic aldehydes and alcohols are common volatiles in flowers [35], the interconversion of alcohols into aldehydes and vice-versa might play a significant role in modulating flower scent and might contribute to attract pollinators. The linear-chained alkanes might also play a significant role in pollinator attraction, besides having a possible function in preventing moisture loss from plant tissues [36,37,38].

Finally, the presence of compounds such as cis-3-hexenyl acetate and methyl salicylate can be reasonably associated with the mechanisms of active plant defense [23,24] in the species *C. laeveipes*.

In conclusion, the chemical composition of the essential oils obtained from the two *Rubiaceae* species, *G. verum* and *C. laevipes*, indicate a complex balance of phytochemicals to protect the plants in their environment. In addition, as shown for other studied Alpine plants [39,40] and plants producing essential oils with similar composition [27,41,42], *G. verum* and *C. laevipes* produce volatiles with valuable biological properties.

## 4. Materials and Methods

### 4.1. Plant Material

*Cruciata laevipes* Opiz and *Galium verum* L. were identified according to Pignatti [3]. Aerial parts were collected at full bloom in the vicinity of Dravugna, Val di Susa, Western Alps (1250 m. asl; N 45°08′47″, E 7°16′46″) in the province of Turin, Italy. Plants were cut at about 1 cm height above ground to avoid soil impurities, samples were weighted and then placed in sealed bottles, half-filled with CH_2_Cl_2_, as a preservative. The *G. verum* flowers were separately collected and stored. Samples were taken to the laboratory within the day and stored at 4 °C, until distillation. Specimens of *C. laevipez* (CL1908) and *G. verum* (GV1935) are deposited at CREA, Lodi, Italy.

### 4.2. Isolation of the Oil

The plant material (about 50 g of the *C. laevipes* whole plant and the *G. verum* leaves and about 35 g of the *G. verum* flowers), to which 0.352 mg of 3-methylcyclohexanone (Sigma-Aldrich (St. Louis, MO, USA), 99% purity) and 0.511 mg of octadecane (Sigma-Aldrich, 99% purity) were added as internal standards, was steam-distilled with odor-free water in a Clevenger-type apparatus, for 1 h. The distillate was saturated with NaCl, extracted with freshly distilled Et_2_O (3 × 100 mL), dried over anhydrous Na_2_SO_4_, and concentrated with a rotary evaporator to give a pale-yellow oil with a yield of 0.01%–0.02%, (weight/fresh weight basis). The resulting oil was then diluted with Et_2_O and analyzed by GC/FID and GC/MS.

### 4.3. Analysis of the Essential Oil

GC/FID analysis was carried out using a Perkin Elmer model 8500 GC (Perkin Elmer Italia Spa, Milano, Italy) equipped with a 30 m × 0.32 mm i.d., Elite-5MS capillary column (0.32 μm film thickness). The sample (0.5 μL) was injected in the “split” mode (1:30), with a column temperature program of 40 °C for 5 min, then increased to 260 °C at 4 °C/min and finally held at that temperature for 10 min. Injector and detector were set at 230 °C and 280 °C, respectively; the carrier gas was He with a head pressure of 12.0 psi.

GC/MS analysis was carried out using a Perkin Elmer Clarus 500 GC equipped with a Clarus 500 mass spectrometer, using the same capillary column and chromatographic conditions as for the GC/FID analysis. Mass spectra were acquired over the 40–500 amu range at 1 scan/sec with ionizing electron energy 70 eV, ion source 230 °C. The transfer line was set at 270 °C, while the carrier gas was He at 1.0 mL/min.

### 4.4. Identification and Quantitation of the Oil Components

The identification of the volatile oil components was performed by their retention indices (AI), their mass spectra, by comparison with the NIST database mass spectral library [43], as well as with literature data [44,45]. Authentic reference compounds purchased from Sigma-Aldrich were also used. Retention indices were calculated using an n-alkane series (C_6_–C_32_) under the same GC conditions as that for the samples. The relative amount of individual components of the oil were expressed as percent peak area relative to total peak area from the GC/FID analysis of the whole extracts. The quantitative data were obtained with GC/FID analysis by the internal standard method, using 3-methylcyclohexanone as the internal reference for compounds with an AI < 1350 (Rt < 25.0 min.; compounds 1–35 in Table 1), and octadecane for compounds with an AI > 1350 (Rt > 25.0 min.; compounds 36–75 in Table 1). A linear proportion between the areas was used, assuming an equal response factor for all detected compounds.

## Figures and Tables

**Figure 1 molecules-25-02333-f001:**
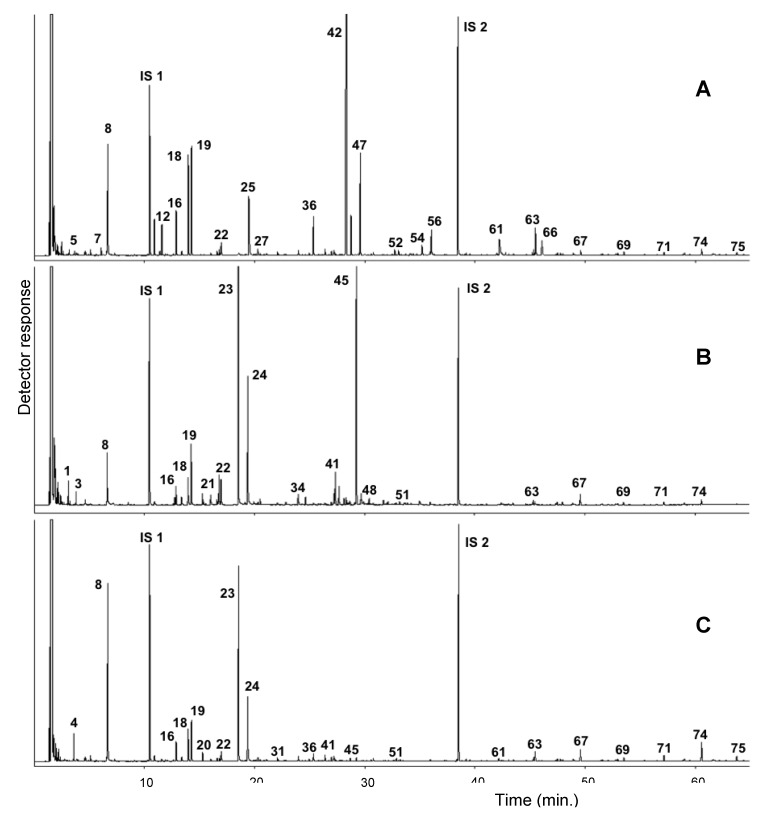
Gaschromatogram of the volatile fractions of *Cruciata laevipez* (**A**) and *Galium verum* flowers (**B**) and leaves (**C**). IS 1: internal standard 1 (3-methylcycohexanone); and IS 2: internal standard 2 (octadecane). For compound identification, see Table 1.

**Figure 2 molecules-25-02333-f002:**
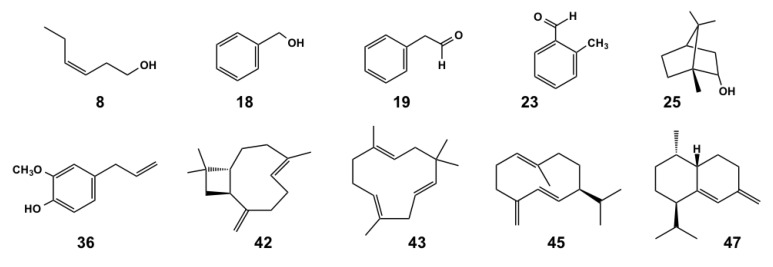
Chemical structure of the most representative compounds detected in the volatile fractions of *Cruciata laevipez* and *Galium verum*. For compound identification, see Table 1.

**Table 1 molecules-25-02333-t001:** Composition (% and μg/g fresh weight) of volatiles from *Galium verum* leaves and flowers and *Cruciata laevipes* whole plant.

				*Galium verum*				*Cruciata laevipez*	
				Leaves		Flowers		Whole Plant	
	Compound ^a^	AI tab ^b^	AI ^c^	%	μg/g	%	μg/g	%	μg/g
**1**	3-Methyl-3-buten-1-ol	731	737	-	-	1.78 ± 0.12	3.96 ± 0.21	0.22 ± 0.03	0.17 ± 0.02
**2**	3-Methyl-1-butanol	740	741	-	-	-	-	0.36 ± 0.06	0.29 ± 0.04
**3**	Pentanol	765	768	tr	tr	1.07 ± 0.19	2.38 ± 0.46	0.33 ± 0.12	0.27 ± 0.09
**4**	*cis*-2-Penten-1-ol	771	775	1.73 ± 0.57	0.75 ± 0.26	tr	tr	0.29 ± 0.08	0.23 ± 0.06
**5**	Hexanal	799	799	0.81 ± 0.26	0.35 ± 0.11	0.52 ± 0.07	1.15 ± 0.17	0.55 ± 0.06	0.44 ± 0.06
**6**	4-Hydroxy-4-methyl pentan-2-one	831	837	-	-	-	-	0.51 ± 0.07	0.41 ± 0.06
**7**	*trans*-2-Hexenal	851	851	tr	tr	-	-	0.47 ± 0.06	0.38 ± 0.06
**8**	*cis*-3-Hexen-1-ol	855	854	17.34 ± 2.41	7.49 ± 0.87	3.35 ± 0.13	7.47 ± 0.40	9.69 ± 1.18	7.84 ± 1.17
**9**	Hexanol	870	869	0.39 ± 0.11	0.17 ± 0.04	2.26 ± 0.10	5.03 ± 0.14	0.27 ± 0.04	0.22 ± 0.04
**10**	Heptanal	904	902	0.10 ± 0.04	0.04 ± 0.02	-	-	0.16 ± 0.02	0.13 ± 0.02
**11**	*trans*-2-Heptenal	958	955	-	-	-	-	0.26 ± 0.01	0.21 ± 0.01
**12**	Benzaldehyde	960	959	1.08 ± 0.06	0.47 ± 0.04	0.30 ± 0.03	0.67 ± 0.08	2.05 ± 0.21	1.66 ± 0.21
**13**	Oct-1-en-3-ol	980	981	0.20 ± 0.02	0.09 ± 0.00	tr	tr	1.92 ± 0.38	1.56 ± 0.35
**14**	6-Methyl-5-hepten-2-ol	992	991	tr	tr	-	-	0.17 ± 0.03	0.14 ± 0.03
**15**	Decane	1000	999	-	-	-	-	0.29 ± 0.02	0.24 ± 0.02
**16**	*cis*-3-Hexenylacetate	1004	1005	3.46 ± 1.30	1.50 ± 0.60	1.12 ± 0.02	2.50 ± 0.04	2.44 ± 0.27	1.97 ± 0.28
**17**	2,4-Heptadienal	1005	1007	0.16 ± 0.02	0.07 ± 0.01	-	-	0.21 ± 0.01	0.17 ± 0.01
**18**	Benzyl alcohol	1042	1034	4.37 ± 0.01	1.89 ± 0.04	1.98 ± 0.04	4.42 ± 0.15	8.30 ± 0.24	6.71 ± 0.24
**19**	Phenylacetaldehyde	1051	1042	5.19 ± 0.62	2.25 ± 0.32	3.43 ± 0.23	7.63 ± 0.62	7.74 ± 0.41	6.26 ± 0.52
**20**	Linalool	1099	1099	0.51 ± 0.07	0.22 ± 0.04	0.29 ± 0.02	0.65 ± 0.05	0.21 ± 0.02	0.17 ± 0.02
**21**	Nonanal	1104	1104	0.38 ± 0.10	0.16 ± 0.04	1.85 ± 0.02	4.12 ± 0.02	0.48 ± 0.04	0.38 ± 0.0.3
**22**	2-Phenylethanol	1106	1110	1.68 ± 0.28	0.73 ± 0.14	1.64 ± 0.08	3.66 ± 0.24	1.16 ± 0.12	0.91 ± 0.09
**23**	2-Methylbenzaldehyde	1152 ^d^	1153	26.27 ± 1.07	11.59 ± 0.73	24.04 ± 1.07	53.54 ± 3.12	0.21 ± 0.02	0.16 ± 0.01
**24**	4-Methylbenzaldehyde	1171 ^d^	1173	7.31 ± 1.49	3.16 ± 0.57	8.45 ± 0.57	18.80 ± 1.01	0.33 ± 0.05	0.26 ± 0.03
**25**	Borneol	1165	1174	-	-	-	-	4.07 ± 0.43	3.21 ± 0.32
**26**	Methyl salicylate	1190	1192	0.30 ± 0.02	0.13 ± 0.01	0.39 ± 0.03	0.87 ± 0.07	0.28 ± 0.02	0.22 ± 0.01
**27**	*α*-Terpineol	1194	1195	0.21 ± 0.03	0.09 ± 0.01	0.12 ± 0.03	0.26 ± 0.06	0.58 ± 0.01	0.46 ± 0.01
**28**	Dodecane	1200	1200	0.44 ± 0.02	0.19 ± 0.01	0.15 ± 0.01	0.33 ± 0.02	0.31 ± 0.01	0.24 ± 0.01
**29**	Decanal	1206	1205	0.27 ± 0.06	0.12 ± 0.03	0.37 ± 0.06	0.83 ± 0.13	0.23 ± 0.02	0.18 ± 0.01
**30**	*β*-Cyclocitral	1217	1222	0.49 ± 0.01	0.21 ± 0.01	0.06 ± 0.01	0.14 ± 0.03	0.23 ± 0.02	0.18 ± 0.02
**31**	Geraniol	1249	1251	0.65 ± 0.04	0.28 ± 0.01	0.22 ± 0.02	0.49 ± 0.03	0.18 ± 0.03	0.14 ± 0.02
**32**	*trans*-2-Decenal	1260	1262	0.04 ± 0.03	0.02 ± 0.02	0.08 ± 0.02	0.17 ± 0.04	0.14 ± 0.01	0.11 ± 0.01
**33**	Indole	1290	1293	0.20 ± 0.02	0.09 ± 0.01	-	-	0.34 ± 0.04	0.27 ± 0.03
**34**	*p*-Vinylguaiacol	1309	1308	0.51 ± 0.29	0.22 ± 0.12	0.85 ± 0.04	1.90 ± 0.07	0.30 ± 0.02	0.24 ± 0.01
**35**	2,4-Decadienal	1315	1318	0.21 ± 0.06	0.09 ± 0.03	0.14 ± 0.01	0.31 ± 0.01	-	-
**36**	Eugenol	1356	1351	1.12 ± 0.09	0.49 ± 0.03	0.20 ± 0.04	0.45 ± 0.09	3.67 ± 0.39	2.89 ± 0.29
**37**	*α*-Copaene	1374	1377	0.14 ± 0.02	0.06 ± 0.01	0.09 ± 0.03	0.20 ± 0.06	0.16 ± 0.02	0.13 ± 0.02
**38**	*β*-Bourbonene	1388	1384	0.93 ± 0.01	0.40 ± 0.01	0.24 ± 0.05	0.53 ± 0.11	0.61 ± 0.04	0.48 ± 0.04
**39**	Isolongifolene	1390	1391	-	-	-	-	0.24 ± 0.03	0.19 ± 0.03
**40**	Tetradecane	1400	1399	0.22 ± 0.05	0.09 ± 0.02	0.13 ± 0.03	0.29 ± 0.07	0.21 ± 0.01	0.16 ± 0.01
**41**	Dodecanal	1408	1408	0.96 ± 0.05	0.42 ± 0.03	3.03 ± 0.32	6.75 ± 0.62	-	-
**42**	*β*-Caryophyllene	1417	1424	0.09 ± 0.03	0.04 ± 0.01	1.33 ± 0.10	2.95 ± 0.18	19.90 ± 2.32	15.68 ± 1.92
**43**	*α*-Humulene	1452	1461	-	-	-	-	2.51 ± 0.04	1.98 ± 0.04
**44**	*allo*-Aromadendrene	1458	1466	-	-	-	-	0.22 ± 0.01	0.17 ± 0.01
**45**	Germacrene D	1484	1475	0.44 ± 0.18	0.19 ± 0.07	27.70 ± 1.67	61.63 ± 2.87	-	-
**46**	*trans*-*β*-Ionone	1487	1480	-	-	-	-	0.15 ± 0.01	0.12 ± 0.01
**47**	*trans*-Muurola-4(14),5-diene	1493	1486	-	-	-	-	7.60 ± 0.42	5.99 ± 0.37
**48**	Bicyclogermacrene	1500	1498	0.26 ± 0.08	0.11 ± 0.04	1.18 ± 0.35	2.63 ± 0.82	-	-
**49**	*cis*-*γ*-Bisabolene	1514	1514	0.10 ± 0.03	0.04 ± 0.01	0.51 ± 0.07	1.14 ± 0.16	-	-
**50**	*δ*-Cadinene	1522	1519	0.27 ± 0.06	0.12 ± 0.03	0.10 ± 0.01	0.22 ± 0.01	0.39 ± 0.02	0.31 ± 0.02
**51**	*trans*-Nerolidol	1561	1560	0.46 ± 0.15	0.20 ± 0.07	0.30 ± 0.03	0.67 ± 0.08	-	-
**52**	C_15_H_22_O MW = 218	-	1579	-	-	-	-	0.70 ± 0.06	0.50 ± 0.05
**53**	C_15_H_24_O MW = 220	-	1588	-	-	-	-	0.79 ± 0.06	0.62 ± 0.04
**54**	C_15_H_24_O MW = 220	-	1640	-	-	-	-	1.01 ± 0.02	0.80 ± 0.01
**55**	C_15_H_24_O MW = 220	-	1644	-	-	-	-	1.20 ± 0.05	0.95 ± 0.04
**56**	Eudesma-4,(15),7-dien-1*β*-ol	1687	1690	-	-	-	-	2.60 ± 0.16	2.05 ± 0.14
**57**	Pentadecanal	1709 ^d^	1705	0.16 ± 0.03	0.07 ± 0.02	0.06 ± 0.01	0.14 ± 0.02	0.19 ± 0.02	0.15 ± 0.02
**58**	Tetradecanoic acid	1764 ^d^	1759	0.33 ± 0.02	0.14 ± 0.01	-	-	0.15 ± 0.02	0.12 ± 0.02
**59**	Hexadecanal	1815 ^d^	1816	0.25 ± 0.09	0.11 ± 0.04	0.36 ± 0.05	0.81 ± 0.13	-	-
**60**	Hexahydrofarnesylacetone	1838 ^d^	1840	0.35 ± 0.13	0.15 ± 0.06	0.08 ± 0.01	0.19 ± 0.02	0.24 ± 0.02	0.19 ± 0.01
**61**	Hexadecanoic acid	1965 ^d^	1961	0.70 ± 0.33	0.31 ± 0.15	0.10 ± 0.01	0.22 ± 0.02	2.52 ± 0.36	1.98 ± 0.27
**62**	Eicosane	2000	1999	0.22 ± 0.04	0.10 ± 0.02	-	-	0.20 ± 0.05	0.16 ± 0.04
**63**	*cis*-Phytol	2079 ^d^	2081	1.72 ± 0.23	0.74 ± 0.12	0.30 ± 0.02	0.66 ± 0.06	2.65 ± 0.65	2.09 ± 0.52
**64**	Heneicosane	2100	2100	0.62 ± 0.10	0.27 ± 0.05	0.34 ± 0.02	0.76 ± 0.05	-	-
**65**	*trans*-Phytol	2121 ^d^	2119	0.36 ± 0.11	0.16 ± 0.05	0.05 ± 0.00	0.11 ± 0.01	0.27 ± 0.03	0.21 ± 0.02
**66**	Linolenic acid	2137 ^d^	2136	0.16 ± 0.13	0.07 ± 0.06	tr	tr	1.60 ± 0.30	1.26 ± 0.24
**67**	Tricosane	2300	2300	1.62 ± 0.17	0.70 ± 0.06	0.71 ± 0.06	1.55 ± 0.16	0.54 ± 0.04	0.43 ± 0.04
**68**	Tetracosane	2400	2400	0.38 ± 0.08	0.17 ± 0.04	0.14 ± 0.02	0.30 ± 0.05	0.23 ± 0.01	0.18 ± 0.01
**69**	Pentacosane	2500	2500	0.59 ± 0.12	0.25 ± 0.05	0.16 ± 0.03	0.36 ± 0.08	0.52 ± 0.08	0.41 ± 0.07
**70**	Hexacosane	2600	2600	0.12 ± 0.03	0.05 ± 0.01	-	-	0.09 ± 0.03	0.07 ± 0.02
**71**	Eptacosane	2700	2700	0.72 ± 0.14	0.31 ± 0.05	0.16 ± 0.03	0.35 ± 0.06	0.37 ± 0.01	0.30 ± 0.01
**72**	Octacosane	2800	2800	-	-	-	-	0.05 ± 0.01	0.04 ± 0.01
**73**	Squalene	2829 ^d^	2828	0.70 ± 0.24	0.31 ± 0.11	-	-	0.20 ± 0.02	0.16 ± 0.02
**74**	Nonacosane	2900	2901	2.98 ± 0.21	1.29 ± 0.12	0.36 ± 0.05	0.80 ± 0.12	0.80 ± 0.05	0.63 ± 0.03
**75**	Entriacontane	3100	3102	0.54 ± 0.23	0.23 ± 0.09	0.04 ± 0.02	0.10 ± 0.05	0.24 ± 0.04	0.19 ± 0.03

	Aldehydes			43.71 ± 0.01	18.91 ± 0.44	42.64 ± 0.46	94.93 ± 2.33	13.02 ± 0.86	10.50 ± 0.95
	Alcohols			25.70 ± 1.67	11.11 ± 0.47	12.09 ± 0.24	26.92 ± 0.89	22.72 ± 1.68	18.35 ± 1.85
	Terpenes			6.85 ± 0.68	2.97 ± 0.36	32.42 ± 1.24	72.15 ± 1.77	46.11 ± 3.21	36.34 ± 2.75
	Hydrocarbons			8.45 ± 0.22	3.66 ± 0.01	2.18 ± 0.25	4.85 ± 0.61	3.87 ± 0.11	3.05 ± 0.10
	Esters			3.76 ± 1.29	1.63 ± 0.59	1.52 ± 0.04	3.37 ± 0.11	2.72 ± 0.29	2.20 ± 0.29
	Phenolics			1.64 ± 0.37	0.71 ± 0.14	1.06 ± 0.09	2.35 ± 0.16	3.97 ± 0.38	3.13 ± 0.28
	Acids			1.19 ± 0.44	0.52 ± 0.21	0.10 ± 0.01	0.22 ± 0.01	4.27 ± 0.04	3.37 ± 0.02
	Miscellaneous			1.04 ± 0.12	0.64 ± 0.20	0.19 ± 0.07	0.43 ± 0.15	1.47 ± 0.09	1.17 ± 0.08
	Total			92.33 ± 0.26	40.13 ± 0.78	92.18 ± 0.41	205.21 ± 1.87	98.14 ± 0.02	78.09 ± 0.62

^a^ Compounds listed in order of elution from an Elite-5 column. ^b^ According to Adams 2006, unless stated otherwise. ^c^ Calculated by GC using n-alkane series (C_6_–C_32_) under the same analytical conditions as for the samples. ^d^ Calculated using authentic reference standards. tr, traces (<0.01%); values >0.01% quoted to nearest 0.01%.

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
