# Peer review of "Identification of the Volatile Components of *Galium verum* L. and *Cruciata leavipes* Opiz from the Western Italian Alps"

_molecules, 2020, doi:10.3390/molecules25102333_

Round 1

Reviewer 1 Report

Reviewer Blind Comments to Authors

Conclusion. The work is interesting, written in understandable language. It brings information to the field about the VOCs in essential oils. Research results are discussed very thoroughly with the literature.

Author Response

Reviewer Blind Comments to Authors

Conclusion. The work is interesting, written in understandable language. It brings information to the field about the VOCs in essential oils. Research results are discussed very thoroughly with the literature.

We are very grateful to Referee #1 for the positive comments on the work.

Reviewer 2 Report

Article entitled "Determination of the volatile components of Galium verum L. and Cruciata leavipes Opiz from the western Italian Alps" is well written and has high scientific relevance, however the following corrections must be made:

1) Improve keywords;

2) Explain in detail, with mathematical equations and discussions, how the quantification by internal standard method was performed.

3) What is the purity of the internal standards used for quantification? Describe in the text.

4) Insert the chemical structures of the most important compounds in the work;

5) Why was it used only 1 hour to isolate the essential oil?

Author Response

We thank Reviewer #2 for helpful and valuable comments. The answer to each comment is reported below.

1) Improve keywords;

Improved. Two additional key words have been inserted.

2) Explain in detail, with mathematical equations and discussions, how the quantification by internal standard method was performed.

As reported (lines 228-232), quantitative evaluation of all the detected compounds was performed by two internal standards: 3-methylcyclohexanone for compounds with an AI < 1350, and octadecane for compounds with an AI > 1350. An equal response factor for all detected compounds was assumed.

In practice, the amount of each compound was obtained by a simple linear proportion between the areas, based on the internal standard area (that is related to a weighted amount) as internal reference. As reported, an equal response factor was assumed (equal to one) for all the detected compounds. This procedure is generally used to quantitate compounds in this type of GC investigation.

Correction has been inserted at line 245.

3) What is the purity of the internal standards used for quantification? Describe in the text.

This information has been inserted in the text at lines 217-218

4) Insert the chemical structures of the most important compounds in the work;

These chemical structures of the most important compounds has now been inserted in a new Figure 2.

5) Why was it used only 1 hour to isolate the essential oil?

We experienced that for this type of herbaceous samples, 1 hour distillation time is more than enough to extract all the volatile compounds. Longer distillation time does not improve the essential oil recovery and composition, it can instead lead to the accumulation of low volatile compounds (e.g. long chain fatty acids) possibly originating by decomposition processes during the distillation, that are not considered as important contributors of the essential oil.

Reviewer 3 Report

Title: Determination of the volatile components of Galium 3 verum L. and Cruciata leavipes Opiz from the western 4 Italian Alps

General comments: This manuscript is focused in an interesting study about the chemical composition of the essential oils produced by two Rubiaceae speacies.

The data on quality, and chemical characterization is interesting, however the authors can improve some points in the present paper.

Thus, I would like to include the following observations:

Title: I suggest the change of Determination from the title to Identification or even Identification and Quantification

Introduction:

Some of the references are too old; authors have the opportunity to change them (1964, 1965, 1974) for others, more recent.

Results:

In the Abstract section, the authors tell that “The chemical composition of the volatile fraction from Galium verum L. (aerial parts and 15 flowers) and Cruciata laevipes Opiz (aerial parts), Rubiaceae, was investigated”

However, in the Results section, (see table 1, please) for Cruciata laevipez the results presented are from the whole plat.

Please, clarify.

Line 37 – “clades” – do you mean classes?

Author Response

We thank Reviewer #3 for helpful and valuable comments. The answer to each comment is reported below.

 General comments: This manuscript is focused in an interesting study about the chemical composition of the essential oils produced by two Rubiaceae speacies.

The data on quality, and chemical characterization is interesting, however the authors can improve some points in the present paper.

Thus, I would like to include the following observations:

Title: I suggest the change of Determination from the title to Identification or even Identification and Quantification

The title has been changed

Introduction:

Some of the references are too old; authors have the opportunity to change them (1964, 1965, 1974) for others, more recent.

We know that some of the reported bibliographic references are old, but the included references are the most appropriate to compare our data. However, to improve bibliography, new references have been considered (39-43) and a new sentence inserted at lines 200-202.

Results:

In the Abstract section, the authors tell that “The chemical composition of the volatile fraction from Galium verum L. (aerial parts and 15 flowers) and Cruciata laevipes Opiz (aerial parts), Rubiaceae, was investigated”

However, in the Results section, (see table 1, please) for Cruciata laevipez the results presented are from the whole plat.

Please, clarify.

The sentence has been corrected referring to Galium verum L. (leaves and flowers) and Cruciata laevipes Opiz (whole plant) in Abstract and in Materials and Methods lines 216-217.

Line 37 – “clades” – do you mean classes?

The term ‘clades’ is correct.

In taxonomy, a clade is defined as a group consisting of a single common ancestor and all descendants of that ancestor.

See also the title of reference 1.

Round 2

Reviewer 2 Report

The article was improved following previous requests.